Gini coefficients for measuring the distribution of sexually transmitted infections among individuals with different levels of sexual activity

Gsteiger Sandro 1
http://orcid.org/0000-0003-4817-8986 Low Nicola 1
Sonnenberg Pam 2
http://orcid.org/0000-0002-4220-5034 Mercer Catherine H. 2
http://orcid.org/0000-0002-5230-6760 Althaus Christian L. 1 christian.althaus@alumni.ethz.ch
1 Institute of Social and Preventive Medicine (ISPM), University of Bern , Bern , Switzerland
2 Institute for Global Health, University College London , London , UK
Zhang Charlie
Electronic publication date: 2020 Jan 20
Publication date: 2020
Volume: 8
Electronic Location ID: e8434
Received 2019 Aug 12; Accepted 2019 Dec 19
Copyright: © 2020 Gsteiger et al.
Copyright year: 2020
Copyright holder: Gsteiger et al.
License: This is an open access article distributed under the terms of the Creative Commons Attribution License, which permits unrestricted use, distribution, reproduction and adaptation in any medium and for any purpose provided that it is properly attributed. For attribution, the original author(s), title, publication source (PeerJ) and either DOI or URL of the article must be cited.
License URL: https://creativecommons.org/licenses/by/4.0/

Keywords: Chlamydia trachomatis, Mycoplasma genitalium, HPV, Sexual behavior, Mathematical model, Transmission model, Gini coefficient, Lorenz curve

Funding: Medical Research Council G0701757 Wellcome Trust 084840 Economic and Social Research Council and Department of Health Swiss National Science Foundation 135654 and 136737 Natsal-3 was supported by grants from the Medical Research Council (G0701757); and the Wellcome Trust (084840); with contributions from the Economic and Social Research Council and Department of Health. Sandro Gsteiger and Christian L. Althaus were funded by the Swiss National Science Foundation (Grants: 135654 and 136737, respectively). The funders had no role in study design, data collection and analysis, decision to publish, or preparation of the manuscript.

==============================
Objectives

Gini coefficients have been used to describe the distribution of Chlamydia trachomatis (CT) infections among individuals with different levels of sexual activity. The objectives of this study were to investigate Gini coefficients for different sexually transmitted infections (STIs), and to determine how STI control interventions might affect the Gini coefficient over time.

Methods

We used population-based data for sexually experienced women from two British National Surveys of Sexual Attitudes and Lifestyles (Natsal-2: 1999–2001; Natsal-3: 2010–2012) to calculate Gini coefficients for CT, Mycoplasma genitalium (MG), and human papillomavirus (HPV) types 6, 11, 16 and 18. We applied bootstrap methods to assess uncertainty and to compare Gini coefficients for different STIs. We then used a mathematical model of STI transmission to study how control interventions affect Gini coefficients.

Results

Gini coefficients for CT and MG were 0.33 (95% CI [0.18–0.49]) and 0.16 (95% CI [0.02–0.36]), respectively. The relatively small coefficient for MG suggests a longer infectious duration compared with CT. The coefficients for HPV types 6, 11, 16 and 18 ranged from 0.15 to 0.38. During the decade between Natsal-2 and Natsal-3, the Gini coefficient for CT did not change. The transmission model shows that higher STI treatment rates are expected to reduce prevalence and increase the Gini coefficient of STIs. In contrast, increased condom use reduces STI prevalence but does not affect the Gini coefficient.

Conclusions

Gini coefficients for STIs can help us to understand the distribution of STIs in the population, according to level of sexual activity, and could be used to inform STI prevention and treatment strategies.

Introduction

Understanding how sexually transmitted infections (STIs) are distributed among individuals is important both from a biological and from a public health perspective. Differences in STI prevalence within a population, between groups with varying levels of sexual activity, can provide information about biological and epidemiological characteristics of the infection. For example, an STI with a long infectious duration, such as human papillomavirus (HPV), will tend to be spread more evenly across a population than an STI with a short infectious duration, such as Neisseria gonorrhoeae (NG). This observation can be explained by the fact that STIs with short infectious durations require a higher rate of sexual partner change for sustained spread in the population. NG is thus more concentrated in a small subgroup of individuals with high sexual activity. Such ideas were initiated in the late 1970s and led to the concept of the “core group” (Hethcote Herbert & Yorke, 1984). In 1990, Brunham & Plummer (1990) inferred the size of core groups for various STIs from the biological parameters that describe transmissibility and infectious duration, and discussed the implications for selecting adequate STI control strategies.

The Gini coefficient can be used to quantify the degree of concentration of an STI in a population. Originally introduced for describing inequalities in income distributions (Gini, 1912), the Gini coefficient provides a general tool to measure the distribution or imbalance of a disease outcome in relation to an exposure variable or risk factor (Lee, 1997), such as the geographic location or sexual behavior. A Gini coefficient of zero denotes perfect equality where an infection is equally distributed across a population. For infections that are concentrated in specific subpopulations, the Gini coefficient can increase up to a maximal value of one. The Lorenz curve is a visual representation of the cumulative distribution of a disease when ordered according to the risk factor (Lorenz, 1905). The diagonal line on a Lorenz curve plot denotes perfect equality, for example, every subpopulation has the same prevalence of an STI.

Several groups have used Gini coefficients and Lorenz curves to describe how Chlamydia trachomatis (CT), NG, syphilis or herpes are distributed across different geographical regions in Canada (Elliott et al., 2002), the UK (Monteiro, Lacey & Merrick, 2005) and the US (Kerani et al., 2005; Chesson et al., 2010a, 2010b). These findings have helped to summarize inequalities in STI distributions, assess the suitability of geographically targeted interventions, and provide insights into the epidemic phases of the STIs over time. Chen, Ghani & Edmunds (2009) were the first to apply the concept of Gini coefficients in mathematical transmission models. They proposed a metapopulation modeling framework that better captures the sociogeographic epidemiology of NG and compared the resulting Gini coefficients with empirical estimates.

The way in which modifiable factors, such as sexual activity and STI control interventions affect the Gini coefficient has been less-well studied. Previously, we described the distribution of CT infections among individuals with different sexual activity using Lorenz curves and Gini coefficients to calibrate dynamic transmission models (Althaus et al., 2012b). We used data from the second British Survey of Sexual Attitudes and Lifestyles (Natsal-2, 1999–2001) (Fenton et al., 2001), which included CT test results (samples were later tested for HPV Johnson et al. (2012)). The most recent survey (Natsal-3, 2010–2012) also provides information on HPV and Mycoplasma genitalium (MG) positivity and offers a unique opportunity to study relationships between sexual behavior and STI prevalence (Sonnenberg et al., 2013, 2015; Tanton et al., 2017).

This study had two objectives. First, we wanted to estimate and compare the Gini coefficients for different STIs and over time using data from two Natsal surveys. Second, we used a mathematical transmission model to obtain general insights into how Gini coefficients are expected to change as a result of STI control interventions.

Methods

Data

Natsal-3 is a population-based probability sample survey of sexual attitudes and lifestyles conducted in Britain (England, Scotland and Wales) and carried out from 2010 to 2012 (Erens et al., 2013; Mercer et al., 2013). The sample consists of 15,162 women and men aged 16–74 years. A subsample of participants aged 16–44 years who reported at least one sexual partner over their lifetime were asked to provide urine samples, resulting in laboratory confirmed STI test results from 2,665 women and 1,885 men (Sonnenberg et al., 2013). Urine was tested for the presence of CT, MG, type-specific HPV, NG and HIV antibodies. To compare Gini coefficients for CT over time, we also used data from the Natsal-2 survey, which was carried out in 1999–2001 (Johnson et al., 2001). This survey includes 11,161 women and men aged 16–44 years. Urine samples for ligase chain reaction testing for CT were available for a subset of 2,055 and 1,474 sexually active women and men aged 18–44 years (Fenton et al., 2001). Unless otherwise stated, we used the subpopulations that provided urine samples for our analysis from both surveys. Sexual behavior and urine sample data were individually weighted to adjust for unequal selection probabilities and to correct for the age, gender and regional profiles in the survey sample. For simplicity, we did not include same-sex contacts for the sexual behavior variables because only 2.2% of the population reported a new same-sex partner during time period covered by Natsal-3. The full datasets of both surveys are available from the UK Data Archive at the University of Essex (http://data-archive.ac.uk).

Statistical analysis

We used Lorenz curves to plot the cumulative proportion of specific STI infections (yi) as a function of the cumulative proportion of the population (xi), after population sub-groups i (i = 1, 2, …, n) have been ranked according to their level of sexual activity. The Gini coefficient is defined as the area between the line of equality and the Lorenz curve over the total area below the line of equality: G=1−∑i=1n⁡(xi−xi−1)(yi+yi−1),

where x0=y0=0 and xn=yn=1.

We derived Lorenz curves and estimated Gini coefficients for CT, MG and HPV types 6, 11, 16 and 18. We focused on these four HPV genotypes because they are present in the widely used quadrivalent vaccine and are frequently considered in dynamic transmission models (Brisson et al., 2016). We did not include NG and HIV in our analysis because of the small numbers of positive tests. We used the number of new opposite-sex partners in the last year as the exposure variable summarizing sexual activity because of its strong association with STI prevalence (Althaus et al., 2012b), and frequent use to parameterize dynamic transmission models (Althaus et al., 2012a). Owing to the larger sample size for female respondents and a potential bias resulting from lower test sensitivity to detect HPV infections in male urine (Sonnenberg et al., 2013), we focused our analysis on women and provide a separate analysis for men as Supplemental Information. We constructed bootstrap confidence intervals (CIs) for the Gini coefficients and point-wise bootstrap confidence bands for the Lorenz curves by sampling with replacement (Kerani et al., 2005; Chesson et al., 2010a, 2010b; Davison & Hinkley, 1997). We calculated the 2.5th and 97.5th percentiles from 1,000 bootstrapped Gini coefficients and Lorenz curves.

Transmission model

We adapted a previously described mathematical model of STI transmission (Althaus et al., 2015) to investigate how changes in infectious duration and transmissibility affect the prevalence and the Gini coefficient of an STI in a simulated population. We stratified the population according to sexual activity (Hethcote Herbert & Yorke, 1984; Garnett & Anderson, 1993), that is, we assumed n different sexual activity classes with 0, 1, 2, …, n−1 new opposite-sex partners per year. For simplicity, we assumed that sexual activity and the natural history and transmission of the infection are the same in women and men. The susceptible-infected-susceptible transmission dynamics can be described by the following ordinary differential equation: dyidt=μ∑j=1n⁡xjyj+βci(1−yi)∑j=1n⁡ρijyj−γyi−μyi,

where yi is the proportion of individuals in sexual activity class i who are infected, and xi is the proportion of all individuals who belong to sexual activity class i. The first and last term of the equation describe how individuals can change their sexual activity class at rate μ and be redistributed to either the same or another sexual activity class proportional to the size of sexual activity classes (Althaus et al., 2015; Fingerhuth et al., 2016). The middle terms describe the process of transmission and the clearance of infection at rate γ. Susceptible individuals 1 − yi have an average of ci new opposite-sex partners per year. β is the per partnership transmission probability and yj is the probability that a partner in sexual activity class j is infected. ρij represents the elements of the sexual mixing matrix (Garnett et al., 1999) ρij=ϵδij+(1−ϵ)cjxj∑l=1nclxl,

where δij denotes the Kronecker delta (it is equal to 1 if i = j and to 0 otherwise).

The proportion of individuals in sexual activity classes xi with ci ∈ 0, 1, 2, …, n − 1 new opposite-sex partners per year were based on Natsal-2. All data for 18–44 year old women and men were pooled and weighted. Since the transmission model is primarily used for illustrative purposes, we did not include changes in sexual behavior, between Natsal-2 and Natsal-3, which were small (Mercer et al., 2013). We set μ = 1 per year as described previously (Althaus et al., 2015; Fingerhuth et al., 2016), and set ∈ = 0, that is, we assumed random proportional mixing between different sexual activity classes (Garnett et al., 1999). We then chose a particular combination of the infectious duration (1/γ) and the per partnership transmission probability (β) and ran the model into endemic equilibrium. We recorded the simulated STI prevalence and computed the Gini coefficient as described above, that is, by calculating the cumulative proportion of infections as a function of the cumulative proportion of the population, ranked by sexual activity i. We repeated this process for various combinations of the infectious duration and the per partnership transmission probability, which allowed us to map between the (unobservable) model parameters and the simulated (observable) summary measures prevalence and Gini coefficient.

We further used the transmission model to investigate how two hypothetical CT control interventions affect infection prevalence and the Gini coefficient. First, we calibrated the infectious duration and the per partnership transmission probability such that the simulated CT prevalence and Gini coefficient correspond to the estimates from Natsal-2 (1/γ = 1.75 years, β = 19%). Second, we simulated the expected changes in prevalence and the Gini coefficient as a result of (a) an increase in CT screening coverage aiming to detect asymptomatic CT cases, and (b) the effects of an educational campaign that leads to a change in sexual behavior and/or an increase in condom use. We assumed that the first intervention would reduce the overall infectious duration by 10%. For the second intervention, we assumed a 10% reduction in the product of the per partnership transmission probability (β) and the number of opposite-sex partners (ci). Note that these hypothetical interventions do not necessarily predict the quantitative effects of real-world interventions; instead, they provide a qualitative picture of how prevalence and Gini coefficients are expected to change.

Data analyses and model simulations were performed in the R software environment for statistical computing (R Core Team, 2016). All code files are available on GitHub (https://github.com/calthaus/gini).

Results

Lorenz curves and Gini coefficients

The Lorenz curves for MG and HPV types 6, 11 and 16 are closer to the diagonal line than the Lorenz curves for CT and HPV 18 (Fig. 1A). This indicates that CT and HPV-18 in women are more strongly associated with the number of new opposite-sex partners in the last year than MG and the other type-specific HPV. The Gini coefficients mirror this observation and are higher for CT (0.33) and HPV 18 (0.38) than for MG and the different HPV types (≤0.22) (Table 1). However, the bootstrapped confidence intervals for the Lorenz curves are wide (Fig. 1B), resulting in considerable uncertainty in the estimated Gini coefficients. The Lorenz curves for CT for the two survey periods of Natsal-2 (1999–2001) and Natsal-3 (2010–2012) are similar (Fig. 1C) with a Gini coefficient in Natsal-2 of 0.30 (95% CI [0.12–0.50]) and in Natsal-3 of 0.33 (95% CI [0.18–0.49]).

Figure 1 Lorenz curves representing the cumulative proportion of STI infections in women as a function of the cumulative proportion of the population, after population sub-groups have been ranked by the number of new opposite-sex partners in the last year.

(A) Lorenz curves for different STIs. Data: Natsal-3. (B) Uncertainty around Lorenz curve for CT. The blue areas represent point-wise 50% (dark blue) and 95% (light blue) confidence bands. Data: Natsal-3. (C) Comparison of Lorenz curves for CT between Natsal-2 (dashed line) and Natsal-3 (solid line). In all graphs, the diagonal line (black dotted line) denotes perfect equality, that is, an equal dispersion of the infection across population sub-groups.

Table 1 Estimated Gini coefficients for different sexually transmitted infections in women.

Infection	Gini coefficient	95% Confidence interval (CI)	
Chlamydia trachomatis	0.33	0.18–0.49	
Mycoplasma genitalium	0.16	0.02–0.36	
HPV 6	0.22	0.02–0.43	
HPV 11	0.17	0.04–0.31	
HPV 16	0.15	0.04–0.27	
HPV 18	0.38	0.10–0.64	

We used the transmission model to perform a mapping between model parameters and the simulated STI prevalence and Gini coefficient. Within the parameter ranges that are representative for the considered STIs, we found a close to linear relationship between the Gini coefficient, infection prevalence, infectious duration and per partnership transmissibility (Fig. 2A, dashed grid). STIs with a short infectious duration (e.g., 1 year) require frequent sexual partner changes (i.e., a high number of opposite-sex partners) and are therefore characterized by a high Gini coefficient. Longer infectious durations increase prevalence, facilitate STI transmission between individuals with a low number of opposite-sex partners, and consequently decrease the Gini coefficient. Interestingly, different values for the per partnership transmission probability only influence prevalence but do not affect the Gini coefficient. These insights from the transmission model potentially allow the inference of biological parameters for the different STIs in Natsal-3 (Fig. 2A, colored dots). Although the confidence intervals and the associated uncertainty are large, CT and HPV 18 seem to be consistent with an infectious duration between 1 and 2 years. MG and the other HPV types are consistent with longer infectious durations.

Figure 2 Relationship between Gini coefficient, STI prevalence, infectious duration and transmissibility.

(A) Gini coefficients and STI prevalence for women in Natsal-3 (colored dots). Modelled values for different combinations of the infectious duration and the per partnership transmission probability are projected on the graph (dashed grid). (B) Expected impact of control measures on Gini coefficients and prevalence of female CT between Natsal-2 and Natsal-3. The black arrows denote a 10% reduction in the per partnership transmission probability (horizontal arrows) or the infectious duration (diagonal arrows).

STI control interventions

We used the transmission model to examine how two hypothetical control interventions for CT affect the Gini coefficient and infection prevalence. First, we assumed that the increase in CT screening between the two survey periods of Natsal-2 (1999–2001) and Natsal-3 (2010–2012) through the National Chlamydia Screening Program in England (Chandra et al., 2017) has resulted in a reduction in the overall infectious duration. The transmission model predicts that this would result in an increase of the Gini coefficient with a concurrent drop in prevalence (Fig. 2B). In contrast, the point estimates of the Gini coefficient and prevalence both show a slight, albeit no statistically significant increase between Natsal-2 and Natsal-3. The expected changes in prevalence and Gini coefficient might be relatively small, however, and may remain within the 95% CIs of the point estimates. Second, we simulated the effects of a decrease in the number of opposite-sex partners and/or an increase in condom. Such a behavior change would be expected to reduce the prevalence of CT without affecting the Gini coefficient.

Discussion

Building upon earlier work (Althaus et al., 2012b), we constructed Lorenz curves and estimated Gini coefficients in women to investigate how different STIs are distributed according to sexual activity. Gini coefficients for CT and HPV 18 appear to be higher than for MG and HPV 6, 11 and 16. We found no evidence that the Gini coefficient for CT changed between the two survey periods of Natsal-2 and Natsal-3. Using a mathematical model of STI transmission, we found that a CT screening intervention should reduce prevalence and increase the Gini coefficient, whilst condom use reduced prevalence but did not affect the Gini coefficient.

A main strength of this study was the availability of Natsal-2 and Natsal-3, two very large data sets that measure both STI positivity and self-reported sexual behavior in probability samples of the British general population. These comprehensive data sets allow comparison between different STIs and over time. Calculating infection prevalence and Gini coefficients is straightforward if suitable data are available. In contrast, obtaining biological parameters that determine the transmission dynamics, such as the infectious duration or the transmission probability, are notoriously difficult to obtain (Althaus et al., 2010, Althaus, Heijne & Low, 2012). In this study, we used a mathematical model of STI transmission with a detailed description of different sexual activity classes (Althaus et al., 2015) to explore the relationship between Gini coefficient, infection prevalence, infectious duration and transmissibility.

There are several limitations to this study. First, despite the large overall sizes of the two Natsal surveys, the relatively low prevalence of STIs in the general population (Fenton et al., 2001; Johnson et al., 2012; Sonnenberg et al., 2013, 2015) resulted in relatively large uncertainties in the Lorenz curves and the corresponding Gini coefficients, particularly for males (see Supplemental Information). Owing to the small sample sizes, we pooled data over all age groups and restricted our analyses to the whole survey population. Investigating sex- or age-specific differences in Gini coefficients would certainly be interesting but is currently not feasible. Second, the limited sample size did not allow us to calculate Gini coefficients for NG and HIV, which have a very low prevalence, and arguably high Gini coefficient (Sonnenberg et al., 2013). Third, the comparison of prevalence and Gini coefficients of CT between Natsal-2 and Natsal-3 should also be treated with caution. The two surveys used different laboratory tests and were not powered to detect changes in CT prevalence (Sonnenberg et al., 2013). Fourth, we defined the exposure variable as the number of new opposite-sex partners in the last year. Using other exposure variables for our analysis would obviously affect the Lorenz curves and the Gini coefficients, but would not necessarily be applicable for the mapping between model parameters and the summary measures. Fifth, mathematical modeling necessarily involves several assumptions and simplifications. As in the data analysis, we did not stratify the population according to age, assumed the sexual behavior in women and men to be the same, and considered the general population of those reporting sex with opposite-sex partners as a whole. Changes in sexual behavior between Natsal-2 and Natsal-3 were minimal (Mercer et al., 2013), so we did not take these into account. We assumed fully proportional mixing which typically results in the best description of infection prevalence in different sexual activity classes in models with a constant per partnership transmission probability (Althaus et al., 2012a, 2015; Garnett et al., 1999). We also did not consider sex-specific differences in the infectious duration and the transmissibility of the various STIs, which might limit the application of Gini coefficients for the inference of these parameters. Further, we assumed that individuals who clear an infection can become reinfected, although CT and type-specific HPV infections might confer temporal immunity (Brunham & Rey-Ladino, 2005; Bogaards et al., 2010). Together, these modeling assumptions highlight that the insights from our conceptual transmission model are primarily qualitative and that quantitative results should be treated with caution.

The calculated Gini coefficients and prevalence of CT and HPV 18 in women in Natsal-3 suggest an infectious duration of 1–2 years, which is in good agreement with previous estimates (Althaus et al., 2010; Insinga et al., 2007; Johnson, Elfström & Edmunds, 2012; Price et al., 2013). Our mapping indicates that the infectious durations for the other high-risk HPV type 16 and low-risk HPV types 6 and 11 could be longer than 2 years. This interpretation would contrast with previous studies that estimate similar infectious durations for HPV-16 and HPV-18 (Insinga et al., 2007; Johnson, Elfström & Edmunds, 2012), and shorter infectious durations of less than a year for HPV 6 and 11 (Insinga et al., 2007). The discrepancy could be a result of ignoring the effects of temporal immunity to reinfection. The infectious duration for the other bacterial STI, MG, seems to be longer than for CT. There is considerable uncertainty regarding the infectious duration of MG. One analysis, which used data from a study of female students in London (UK), estimated the mean infectious duration at 15 months (Smieszek & White, 2016), which is in the same range as CT. The per partnership transmission probability is a highly model-dependent parameter and depends on the type of sexual partnerships that are considered. It is maybe not surprising that our mapping, which suggests relatively low per partnership transmission probabilities of 10–25%, is not consistent with estimates from other modeling studies (Althaus, Heijne & Low, 2012; Bogaards et al., 2010).

Our findings allowed us to interpret differences in Gini coefficients and changes over time for different STIs, illustrating that Gini coefficients can serve beyond their original role as simple statistical measures of exposure-outcome associations. We found that changes in the transmission probability only influence infection prevalence. This means that while decreasing the transmission probability (e.g., through increased condom use) decreases the overall burden of an STI, it does not affect how an STI is distributed among individuals with different sexual activity. Hence, the target groups for future control interventions should remain the same. In contrast, we showed that changes in the infectious duration (e.g., through increased testing and treatment uptake) affects both the prevalence and Gini coefficient of an STI. A stronger concentration of the infections among individuals with increased sexual activity would require a change in the target groups for control interventions. A prerequisite for this newly proposed use of Gini coefficients for STIs will be the availability of large data sets with a sufficiently high infection prevalence. These could either be population-based surveys or smaller cohorts that focus on individuals with an increased risk of acquiring STIs, such as men who have sex with men.

Conclusion

In summary, our study illustrates that the Gini coefficient for measuring the distribution of an STI among individuals with different sexual activity represents a simple proxy measure, which combines epidemiological and behavioral data. Estimating Gini coefficients for the general population or particular sub-populations, in combination with mathematical modeling, has the potential to make inference of biological parameters that determine STI transmission and to assess the impact of control measures.

Supplemental Information

Supplemental Information 1 Additional analysis for men.

Click here for additional data file.

Natsal-3 is a collaboration between University College London (UCL), London School of Hygiene and Tropical Medicine (LSHTM), National Centre for Social Research (NatCen), Public Health England (PHE), and the University of Manchester. Natsal-2 was a collaboration between UCL, LSHTM, NatCen and the Health Protection Agency (now PHE). We thank the study participants, the team of interviewers from NatCen Social Research, and operations and computing staff from NatCen Social Research. Natsal-3 was approved by the Oxfordshire Research Ethics Committee A (Ref: 10/H0604/27); Natsal-2 was approved by the University College Hospital and North Thames Multi-Centre Research Ethics Committee and all the Local Research Ethics Committees in Britain.

Additional Information and Declarations

Competing Interests

Author Contributions

Data Availability

Christian L. Althaus is an Academic Editor for PeerJ. All other authors declare no competing interests.

Sandro Gsteiger conceived and designed the experiments, performed the experiments, analyzed the data, prepared figures and/or tables, authored or reviewed drafts of the paper, and approved the final draft.

Nicola Low conceived and designed the experiments, analyzed the data, authored or reviewed drafts of the paper, and approved the final draft.

Pam Sonnenberg analyzed the data, authored or reviewed drafts of the paper, and approved the final draft.

Catherine H. Mercer analyzed the data, authored or reviewed drafts of the paper, and approved the final draft.

Christian L. Althaus conceived and designed the experiments, performed the experiments, analyzed the data, prepared figures and/or tables, authored or reviewed drafts of the paper, and approved the final draft.

The following information was supplied regarding data availability:

All data and R code files are available on the following GitHub repository: https://github.com/calthaus/gini.

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
