# Peer review of "Gini coefficients for measuring the distribution of sexually transmitted infections among individuals with different levels of sexual activity"

_PeerJ, doi:10.7717/peerj.8434_

## Round 0.1 · original submission · Major Revisions

I concur with the reviewers' critics and suggestions, especially those made by Reviewer 2 about providing detailed interpretations of Gini coefficient measures used in this manuscript.

Reviewer 1 ·

Basic reporting

The paper is clearly written, and well referenced. The paper is also well-structured, and assumptions and results are well described and presented.

Experimental design

The research question and objectives are well posed and clearly addressed. The analyses seemed to have been performed rigorously, and the right data was used.

Validity of the findings

Although the general idea (using Gini coefficient to make inference of biological parameters that determine STI transmission and to assess the impact of control measures) is interesting and potentially useful, the data used is still limited and therefore the uncertainties too large to make any strong conclusions, which has been stated as a limitation by the authors. Perhaps the authors should suggest/describe what data would be needed to be able to make stronger predictions/conclusions.

Additional comments

I think this is an interesting paper that makes a clear contribution to the field, thus I support its publication. However, I left a few comments throughout the paper for the authors to consider.

Annotated reviews are not available for download in order to protect the identity of reviewers who chose to remain anonymous.

·

Basic reporting

The document is consistent with the basic reporting guidelines, with the exceptions listed below.
The language of 80:83 could be improved for clarity and substance. Specifically (in order of importance):
1. Misc: It is not generally clear when the comparison of indices is in time, or when it is being compared to the ODE data.
2. 80:81 it would be helpful to clarify the comparison over time.
3. 82:83 is too long and the intent is unclear.
4. References in 163:167 (discussed later as well)

Experimental design

1. Raw data showing the simulated Gini index and the true Gini index have not been provided.
2. It is not clear how and where the Gini index is used to correlate data from the ODE.
3. The Gini index as an indicator of imbalance in the data set has not been addressed clearly. That a higher Gini index shows imbalance should clearly be explained.

Validity of the findings

1. 165:166 The change in the model parameter, and subsequent change in output has been linked to education and condom usage in an unsubstantiated manner. It is imperative to discuss the parameters of a campaign which would cause such a change in the model parameter. (Lee WC. [4]) describes a clearer example of simulating a parameter change.
2. 163:165 is also unsubstantiated.
3. The interpretation and understanding which is to be drawn from 209:210 (point estimates) is not clearly explained.
4. No data has been provided to describe how changes in the ODE model are able to provide insights to the Gini coefficient
5. 211:215 as noted previously, the changes need to be explained better. How is the transmission probability factored into the Gini coefficient?

Additional comments

1. The analysis might be improved by incorporating the interpretation of the Gini index given by Rogerson, P.A. Lett Spat Resour Sci (2013) 6: 109. (https://doi.org/10.1007/s12076-013-0091-x).
2. Natsal-2 is not mentioned in the acknowledgements, though Natsal-3 has been mentioned.
3. It would be nice, but not necessary to have a plot showing the Lorentz curves

I am convinced that the authors have acted in good faith and commend their well commented R scripts on Github. I am certain that after the revisions requested have been addressed satisfactorily, the paper should be Accepted.

---

## Round 0.2 · accepted · Accept

We are grateful to you for contributing your manuscript to PeerJ and hope to receive your high quality submissions in the future.

Reviewer 1 ·

Basic reporting

OK

Experimental design

OK

Validity of the findings

OK

Additional comments

None

·

Basic reporting

No comments.

Experimental design

No Comments.

Validity of the findings

No comments.

Additional comments

The authors have expanded the manuscript in response to the previous reviews in a commendable manner. The work presented, on the introduction and usage of the Gini coefficient for disease transmission is now clear. The authors have clarified the aims and the study is reproducible (with well documented code), and is a worthy contribution to the existing literature. The simulated usage of the Gini coefficient as a predictive tool has also been described well.